# PITFALLS OF GAUSSIANS AS A NOISE DISTRIBUTION IN NCE

**Holden Lee**[*]
Johns Hopkins University
hlee283@jhu.edu

**Chirag Pabbaraju**[*]
Stanford University
cpabbara@cs.stanford.edu

**Anish Sevekari**[*]
Carnegie Mellon University
asevekar@andrew.cmu.edu

**Andrej Risteski**
Carnegie Mellon University
aristesk@andrew.cmu.edu

## ABSTRACT

Noise Contrastive Estimation (NCE) is a popular approach for learning probability density functions parameterized up to a constant of proportionality. The main idea is to design a classification problem for distinguishing training data from samples from an easy-to-sample noise distribution $q$, in a manner that avoids having to calculate a partition function. It is well-known that the choice of $q$ can severely impact the computational and statistical efficiency of NCE. In practice, a common choice for $q$ is a Gaussian which matches the mean and covariance of the data. In this paper, we show that such a choice can result in an exponentially bad (in the ambient dimension) conditioning of the Hessian of the loss, even for very simple data distributions. As a consequence, both the statistical and algorithmic complexity for such a choice of $q$ will be problematic in practice, suggesting that more complex and tailored noise distributions are essential to the success of NCE.

## 1 INTRODUCTION

Noise contrastive estimation (NCE), introduced in (Gutmann & Hyvärinen, 2010; 2012), is one of several popular approaches for learning probability density functions parameterized up to a constant of proportionality, i.e. $p(x) \propto \exp(E_\theta(x))$, for some parametric family $\{E_\theta\}_\theta$. A recent incarnation of this paradigm is, for example, energy-based models (EBMs), which have achieved near-state-of-the-art results on many image generation tasks (Du & Mordatch, 2019; Song & Ermon, 2019). The main idea in NCE is to set up a self-supervised learning (SSL) task, in which we train a classifier to distinguish between samples from the data distribution $P_*$ and a known, easy-to-sample distribution $Q$, often called the "noise" or "contrast" distribution. It can be shown that for a large choice of losses for the classification problem, the optimal classifier model is a (simple) function of the density ratio $p_*/q$, so an estimate for $p_*$ can be extracted from a good classifier. Moreover, this strategy can be implemented *while avoiding* calculation of the partition function, which is necessary when using maximum likelihood to learn $p^*$.

The noise distribution $q$ is the most significant "hyperparameter" in NCE training, with both strong empirical (Rhodes et al., 2020) and theoretical (Liu et al., 2021) evidence that a poor choice of $q$ can result in poor algorithmic behavior. (Chehab et al., 2022) show that even the optimal $q$ for finite number of samples can have an unexpected form (e.g., it is not equal to the true data distribution $p_*$). Since $q$ needs to be a distribution that one can efficiently draw samples from, as well as write an expression for the probability density function, the choices are somewhat limited.

A particularly common way to pick $q$ is as a Gaussian that matches the mean and covariance of the input data (Gutmann & Hyvärinen, 2012; Rhodes et al., 2020). Our main contribution in this paper is to formally show that such a choice can result in an objective that is statistically poorly behaved, even for relatively simple data distributions. We show that even if $p^*$ is a *product distribution* and a member of a very simple *exponential family*, the Hessian of the NCE loss, when using a Gaussian

---

[*]Alphabetical ordering.

noise distribution $q$ with matching mean and covariance has exponentially small (in the ambient dimension) spectral norm. As a consequence, the optimization landscape around the optimum will be exponentially flat, making gradient-based optimization challenging. As the main result of the paper, we show the asymptotic sample efficiency of the NCE objective will be *exponentially bad* in the ambient dimension.

## 2 OVERVIEW OF RESULTS

Let $P_*$ be a distribution in a parametric family $\{P_\theta\}_{\theta \in \Theta}$. We wish to estimate $P_*$ via $P_\theta$ for some $\theta_* \in \Theta$ by solving a noise contrastive estimation task. To set up the task, we also need to choose a noise distribution $Q$, with the constraint that we can draw samples from it efficiently, and we can evaluate the probability density function efficiently. We will use $p_\theta, p_*, q$ to denote the probability density functions (pdfs) of $P_\theta$, $P_*$, and $Q$. For a data distribution $P_*$ and noise distribution $Q$, the NCE loss of a distribution $P_\theta$ is defined as follows:

**Definition 1** (NCE Loss). The NCE loss of $P_\theta$ w.r.t. data distribution $P_*$ and noise $Q$ is

$$L(P_\theta) = -\frac{1}{2}\mathbb{E}_{P_*} \log \frac{p_\theta}{p_\theta + q} - \frac{1}{2}\mathbb{E}_Q \log \frac{q}{p_\theta + q}. \tag{1}$$

Moreover, the empirical version of the NCE loss when given i.i.d. samples $(x_1, \ldots, x_n) \sim P_*^n$ and $(y_1, \ldots, y_n) \sim Q^n$ is given by

$$L^n(\theta) = \frac{1}{n}\sum_{i=1}^{n} -\frac{1}{2}\log \frac{p_\theta(x_i)}{p_\theta(x_i) + q(x_i)} + \frac{1}{n}\sum_{i=1}^{n} -\frac{1}{2}\log \frac{q(y_i)}{p_\theta(y_i) + q(y_i)}. \tag{2}$$

By a slight abuse of notation, we will use $L(\theta), L(p_\theta)$ and $L(P_\theta)$ interchangeably.

The NCE loss can be interpreted as the binary cross-entropy loss for the classification task of distinguishing the data samples from the noise samples. To avoid calculating the partition function, one considers it as an additional parameter, namely we consider an augmented vector of parameters $\tilde{\theta} = (\theta, c)$ and let $p_{\tilde{\theta}}(x) = \exp(E_\theta(x) - c)$. The crucial property of the NCE loss is that it has a unique minimizer:

**Lemma 2** (Gutmann & Hyvärinen 2012). *The NCE objective in Definition 1 is uniquely minimized at $\theta = \theta_*$ and $c = \log(\int_x \exp(E_{\theta^*}(x))dx)$ provided that the support of $Q$ contains that of $P_*$.*

We will be focusing on the Hessian of the loss $L$, as the crucial object governing both the algorithmic and statistical difficulty of the resulting objective. We will show the following two main results:

**Theorem 3** (Exponentially flat Hessian). *For $d > 0$ large enough, there exists a distribution $P_* = P_{\theta_*}$ over $\mathbb{R}^d$ such that*

- $\mathbb{E}_{P_*}[x] = 0$ *and* $\mathbb{E}_{P_*}[xx^\top] = I_d$.
- $P_*$ *is a product distribution, namely* $p_*(x_1, x_2, \ldots, x_d) = \prod_{i=1}^{d} p^*(x_i)$.
- *The NCE loss when using* $q = \mathcal{N}(0, I_d)$ *as the noise distribution has the property that*

$$\|\nabla^2 L(\theta_*)\|_2 \leq \exp\left(-\Omega(d)\right).$$

We remark the above example of a problematic distribution $P^*$ is extremely simple. Namely, $P^*$ is a product distribution, with 0 mean and identity covariance. It actually is also the case that $P^*$ is log-concave—which is typically thought of as an "easy" class of distributions to learn due to the fact that log-concave distributions are unimodal.

The fact that the Hessian is exponentially flat near the optimum means that gradient-descent based optimization without additional tricks (e.g., gradient normalization, second order methods like Newton's method) will fail. (See, e.g., Theorem 4.1 and 4.2 in Liu et al. (2021).) For us, this will be merely an intermediate result. We will address a more fundamental issue of the sample complexity of NCE, which is independent of the optimization algorithm used. Namely, we will show that without a large number of samples, the best minimizer of the empirical NCE might not be close to the target distribution. Proving this will require the development of some technical machinery.

More precisely, we use the result above to show that the asymptotic statistical complexity, using the above choice of $P^*, Q$, is exponentially bad in the dimension. This substantially clarifies results in

Gutmann & Hyvärinen (2012), who provide an expression for the asymptotic statistical complexity in terms of $P^*, Q$ (Theorem 3, Gutmann & Hyvärinen (2012)), but from which it's very difficult to glean quantitatively how bad the dependence on dimension can be for a particular choice of $P^*, Q$. Unlike the landscape issues that (Liu et al., 2021) point out, the statistical issues are impossible to fix with a better optimization algorithm: they are fundamental limitations of the NCE loss.

**Theorem 4** (Asymptotic Statistical Complexity). *Let $d > 0$ be sufficiently large and $Q = \mathcal{N}(0, I_d)$. Let $\hat{\theta}_n$ be the optimizer for the empirical NCE loss $L^n(\theta)$ with the data distribution $P_*$ given by Theorem 3 above and noise distribution $Q$. Then, as $n \to \infty$, the mean-squared error satisfies*

$$\mathbb{E}\left[\left\|\hat{\theta}_n - \theta_*\right\|_2^2\right] = \frac{\exp(\Omega(d))}{n}.$$

## 3 EXPONENTIALLY FLAT HESSIAN: PROOF OF THEOREM 3

The proof of Theorem 3 consists of three ingredients. First, in Section 3.1, we will compute an algebraically convenient upper bound for the spectral norm of the Hessian of the loss (eq. (1)). We will restrict our attention to the case when $\{P_\theta\}$ belongs to an exponential family. The upper bound will be in terms of the total variation distance $\text{TV}(P_*, Q)$ and the Fisher information matrix of the sufficient statistics at $\theta_*$. Here, $P_*$ denotes the true data distribution and $Q$ denotes the noise distribution.

Then, in Section 3.2, we construct a distribution $P^*$ for which the TV distance between $P^*$ and $Q$ is large. We do this by "tensorizing" a univariate distribution. Namely, we construct a univariate distribution with mean 0 and variance 1 that is at a constant TV distance from a standard univariate Gaussian. Then, we use the fact that the *Hellinger* distance tensorizes, along with the relationship between TV and Hellinger distance, to show that $TV(P^*, Q) \geq 1 - \delta^d$ for some constant $\delta < 1$. (See Wasserman (2020) for a detailed review of distance measures.) Section 3.3 bounds the Fisher information matrix term, completing all the components required to establish Theorem 3.

### 3.1 BOUNDING THE HESSIAN IN TERMS OF TV DISTANCE

Suppose $\{P_\theta\}$ is an exponential family of distributions, that is $p_\theta(x) = \exp(\theta^\top T(x))$, where $T(x)$ is a known function. Then, a straightforward calculation (see e.g., Appendix A in Liu et al. (2021)) shows that the gradient and the Hessian of the NCE loss (eq. (1)) with respect to $\theta$ have the following forms:

$$\nabla_\theta p_\theta(x) = p_\theta(x) \cdot T(x), \tag{3}$$

$$\nabla_\theta L(p_\theta) = \frac{1}{2} \int_x \frac{q}{p_\theta + q}(p_\theta - p_*)T(x)dx, \tag{4}$$

$$\nabla_\theta^2 L(p_\theta) = \frac{1}{2} \int_x \frac{(p_* + q)p_\theta q}{(p_\theta + q)^2}T(x)T(x)^\top dx. \tag{5}$$

For $\theta = \theta_*$ and $p_\theta = p_*$, we have

$$\nabla_\theta^2 L(p_{\theta_*}) = \frac{1}{2} \int_x \frac{p_* q}{p_* + q}T(x)T(x)^\top dx \preceq \frac{1}{2} \int_x \min(p_*, q)T(x)T(x)^\top dx \tag{6}$$

The second line holds since $\frac{p_* q}{p_* + q} = \min(p_*, q) \cdot \frac{\max(p_*, q)}{p_* + q} \leq \min(p_*, q)$. Applying the matrix version of the Cauchy-Schwarz inequality (Lemma 9, Appendix A) to eq. (6) with two parts $\frac{\min(p_*(x), q(x))}{\sqrt{p_*(x)}}$ and $T(x)T(x)^\top \sqrt{p_*(x)}$, we obtain

$$\|\nabla_\theta^2 L(P_*)\|_2 \leq \|\nabla_\theta^2 L(P_*)\|_F \leq \frac{1}{2}\left(\int_x \frac{\min(p_*, q)^2}{p_*}\right)^{\frac{1}{2}}\left(\int_x \|T(x)T(x)^\top\|_F^2 p_*(x)dx\right)^{\frac{1}{2}}$$

$$\leq \frac{1}{2}\left(\int_x \min(p_*, q)dx\right)^{\frac{1}{2}}\left(\int_x \|T(x)T(x)^\top\|_F^2 p_*(x)dx\right)^{\frac{1}{2}}$$

$$\implies \|\nabla_\theta^2 L(P_*)\|_2 \leq \frac{1}{2}\left(1 - \text{TV}(P_*, Q)\right)^{\frac{1}{2}}\left(\int_x \|T(x)T(x)^\top\|_F^2 p_*(x)dx\right)^{\frac{1}{2}}. \tag{7}$$

We bound the two terms in the product above separately. The first term is small when $P_*$ and $Q$ are significantly different. The second term is an upper bound of the Frobenius norm of the Fisher matrix at $P_*$. We will construct $P_*$ such that the first term dominates, giving us the upper bound required.

## 3.2 Constructing the hard distribution $P_*$

The hard distribution $P_*$ over $\mathbb{R}^d$ will have the property that $\mathbb{E}_{P_*}[x] = 0$, $\mathbb{E}_{P_*}[xx^\top] = I_d$, but will still have large TV distance from the standard Gaussian $Q = \mathcal{N}(0, I_d)$. This distribution will simply be a product distribution—the following lemma formalizes our main trick of tensorization to construct a distribution having large TV distance with the Gaussian.

**Lemma 5.** *Let $d > 0$ be given. Let $Q = \mathcal{N}(0, I_d)$ be the standard Gaussian in $\mathbb{R}^d$. Then, for some $\delta < 1$, there exists a log-concave distribution $P$ (also over $\mathbb{R}^d$) with mean 0 and covariance $I_d$ satisfying $\mathrm{TV}(P, Q) \geq 1 - \delta^d$.*

*Proof.* Let $\hat{Q}$ denote the standard normal distribution over $\mathbb{R}$. Let $\hat{P}$ be any other distribution over $\mathbb{R}$ with mean 0 and variance 1 that satisfies $\rho(\hat{P}, \hat{Q}) = \delta < 1$, where $\rho(\hat{P}, \hat{Q}) = \int_x \sqrt{\hat{p}\hat{q}}\, dx$ is the Bhattacharya coefficient. Since $\rho$ tensorizes (Wasserman, 2020), we have that $\rho(\hat{P}^d, \hat{Q}^d) = \rho(\hat{P}, \hat{Q})^d$ for any $d > 1$. We can then write the Hellinger distance between $P, Q$ as

$$H^2(P, Q) := 1 - \int_x \sqrt{pq}\, dx = 2(1 - \rho(\hat{P}, \hat{Q})^d). \tag{8}$$

Further, we also know that

$$\frac{1}{2}H^2(\hat{P}^d, \hat{Q}^d) \leq \mathrm{TV}(\hat{P}^d, \hat{Q}^d) \implies 1 - \rho(\hat{P}, \hat{Q})^d \leq \mathrm{TV}(\hat{P}^d, \hat{Q}^d) \implies 1 - \delta^d \leq \mathrm{TV}(\hat{P}^d, \hat{Q}^d).$$

Setting $P = \hat{P}^d$ and noting that $\hat{Q}^d = Q = \mathcal{N}(0, I_d)$, we have $\mathrm{TV}(P, Q) \geq 1 - \delta^d$. Finally, if the chosen $\hat{P}$ is a log-concave distribution, then so is $\hat{P}^d$, since the product of log-concave distributions is log-concave, which completes the proof. $\qquad\square$

We will now explicitly define the distribution $P_*$ that we will work with for rest of the paper.

**Definition 6.** Consider the exponential family $\big\{p_\theta(x) = \exp\big(\theta^\top T(x)\big)\big\}_{\theta \in \mathbb{R}^{d+1}}$ given by the sufficient statistics $T(x) = (x_1^4, \ldots, x_d^4, 1)$. Let $P_* = \hat{P}^d$ where $\hat{P}$ is the distribution on $\mathbb{R}$ with density function $\hat{p}$ given by

$$\hat{p}(x) \propto \exp\left(-\frac{x^4}{\sigma^4}\right).$$

We will set the constant of proportionality $C$ and $\sigma$ appropriately to ensure that $\hat{P}$ has mean 0 and variance 1. Note that $P_* = P_{\theta_*}$ for $\theta_* = -\big(\frac{1}{\sigma^4}, \ldots, \frac{1}{\sigma^4}, \log C\big)$.

Since $\frac{d^2 \log \hat{p}}{dx^2} = -\frac{12x^2}{\sigma^4} \leq 0$, $\hat{p}$ is log-concave. Further, symmetry of $\hat{p}$ around the origin gives $\mathbb{E}[\hat{P}] = 0$, and the choice of $\sigma$ ensures that $\mathrm{Var}[\hat{P}] = 1$. The normalizing constant $C$ satisfies

$$C = \int_{-\infty}^{\infty} e^{-\frac{x^4}{\sigma^4}}\, dx = 2\int_0^\infty e^{-\frac{x^4}{\sigma^4}}\, dx.$$

Substituting $t = \frac{x^4}{\sigma^4}$, $dt = \frac{4x^3}{\sigma^4}dx = \frac{4t^{3/4}}{\sigma}dx$ gives

$$C = \frac{\sigma}{2}\int_0^\infty t^{-3/4}e^{-t}dt = \frac{\sigma}{2}\Gamma\left(\frac{1}{4}\right) = 2\sigma\Gamma\left(\frac{5}{4}\right).$$

where $\Gamma(z)$ is the gamma function defined as $\Gamma(z)\int_0^\infty x^{z-1}e^{-x}dx$. The variance is given by

$$\mathrm{Var}\left[\hat{P}\right] = \frac{1}{C}\int_{-\infty}^\infty x^2 e^{-\frac{x^4}{\sigma^4}}\, dx = \frac{2}{C}\int_0^\infty x^2 e^{-\frac{x^4}{\sigma^4}}\, dx.$$

The same substitution as above gives

$$\mathrm{Var}(\hat{P}) = \frac{1}{2C}\int_0^\infty t^{1/2}t^{-3/4}\sigma^3 e^{-t}dt = \frac{\sigma^3}{2C}\int_0^\infty t^{-1/4}e^{-t}dt = \frac{\sigma^3}{2C}\Gamma\left(\frac{3}{4}\right) = \frac{\sigma^2}{4}\frac{\Gamma(3/4)}{\Gamma(5/4)}.$$

Thus, setting $\sigma = \sqrt{\frac{4\Gamma(5/4)}{\Gamma(3/4)}}$ results in $\text{Var}[\hat{P}] = 1$. Correspondingly, we have $C = \frac{4\Gamma(5/4)^{3/2}}{\sqrt{\Gamma(3/4)}}$.

For this choice of $\hat{P}$, the Bhattacharya coefficient $\rho(\hat{P}, \hat{Q})$ is given by:

$$\rho(\hat{P}, \hat{Q}) = \int_{-\infty}^{\infty} \sqrt{\hat{p}(x)\hat{q}(x)}dx = \frac{1}{\sqrt{C\sqrt{2\pi}}} \int_{-\infty}^{\infty} \exp\left(-\frac{x^2}{4} - \frac{x^4}{2\sigma^4}\right) dx \approx 0.9905 \leq 0.991 < 1.$$

Thus, in the proof of Lemma 5, we can use this choice of $\hat{P}$, and we have that for $\delta = 0.991$ and $P_* = \hat{P}^d$, $\text{TV}(P_*, Q) \geq 1 - \delta^d$, as required.

### 3.3 BOUNDING THE FISHER INFORMATION MATRIX

In this subsection, we bound the second factor in eq. (7), which is an upper bound on the Frobenius norm of the Fisher information matrix at $\theta_*$.

**Lemma 7.** *For some constant $M > 0$, we have*

$$\int_x \left\|T(x)T(x)^\top\right\|_F^2 p_*(x)dx \leq d^2 M, \tag{9}$$

*Proof.* Recall that $T(x) = (x_1^4, \ldots, x_d^4, 1)$. Then,

$$\left\|T(x)T(x)^\top\right\|_F^2 = \sum_i x_i^{16} + \sum_{i \neq j} x_i^8 x_j^8 + 2\sum_i x_i^4 + 1. \tag{10}$$

Therefore, by linearity of expectation, and using the fact that $P_*$ is a product distribution,

$$\int_x \left\|T(x)T(x)^\top\right\|_F^2 p_*(x)dx = d \cdot \mathbb{E}_{\hat{P}}[x^{16}] + d(d-1) \cdot \left(\mathbb{E}_{\hat{P}}[x^8]\right)^2 + 2d \cdot \mathbb{E}_{\hat{P}}[x^4] + 1 \leq d^2 M,$$

for an appropriate choice of constant $M$. This constant exists since all the expectations above are bounded owing to the fact that the exponential density $\hat{p}$ dominates in the integrals. $\qquad \square$

### 3.4 PUTTING THINGS TOGETHER

For $P_*$ defined as above, and $Q = \mathcal{N}(0, I_d)$, Lemma 5 ensures that $1 - \text{TV}(P_*, Q) \leq \delta^d$, for $\delta = 0.991$. From Lemma 7, we have that

$$\int_x \left\|T(x)T(x)^T\right\|_F^2 p_*(x)dx \leq d^2 M.$$

Substituting these bounds in eq. (7), we get that

$$\left\|\nabla_\theta^2 L(P_*)\right\|_2 \leq \frac{1}{2}\delta^{d/2} d\sqrt{M} = \exp(-\Omega(d)).$$

By construction, $p_*$ is a product distribution with $\mathbb{E}_{p_*}[x] = 0$ and $\mathbb{E}_{p_*}[xx^\top] = I_d$, which completes the proof of the theorem.

## 4 PROOF OF THEOREM 4

We will bound the error of the optimizer $\hat{\theta}_n$ of the empirical NCE loss (eq. (2)) using the bias-variance decomposition of MSE. To do this, we will reason about the random variable $\sqrt{n}(\hat{\theta}_n - \theta_*)$; let $\Sigma$ be its covariance matrix. Since $\hat{\theta}_n$ is an unbiased estimate of $\theta_*$, the MSE decomposes as

$$\mathbb{E}\left[\left\|\hat{\theta}_n - \theta_*\right\|_2^2\right] = \frac{1}{n}\text{Tr}(\Sigma). \tag{11}$$

The proof of Theorem 4 proceeds as follows. In Section 4.1, we show that the random variable $\sqrt{n}(\hat{\theta}_n - \theta_*)$ is asymptotically normal with mean 0 and covariance matrix $\Sigma$ given by

$$\Sigma = \nabla_\theta^2 L(\theta_*)^{-1}\text{Var}\left[\sqrt{n}\nabla_\theta L^n(\theta_*)\right]\nabla_\theta^2 L(\theta_*)^{-1}. \tag{12}$$

We prove that the Hessian $\nabla_\theta^2 L(\theta_*)$ is invertible in Appendix C, so that the above expression is well-defined. Since $\Sigma \succeq 0$ (it is a covariance matrix), to get a lower bound on $\text{Tr}(\Sigma)$, it suffices to get a lower bound on the largest eigenvalue of $\Sigma$. Looking at the factors on the right hand side of eq. (12), we note first that Theorem 3 ensures an exponential lower bound on *all* eigenvalues of $\nabla_\theta^2 L(\theta_*)^{-1}$. The bulk of the proof towards lower bounding the largest eigenvalue of $\Sigma$ consists of lower bounding $\text{Var}\big[v^\top \cdot \sqrt{n}\nabla_\theta L^n(\theta_*)\big]$, the *directional* variance of $\sqrt{n}\nabla_\theta L^n(\theta_*)$ along a suitably chosen direction $v$ in terms of $v^\top \nabla_\theta^2 L(\theta_*)v$. In Section 4.2 and Section 4.3, we use anti-concentration bounds to prove such variance lower bounds.

## 4.1 GAUSSIAN LIMIT OF $\sqrt{n}(\hat{\theta}_n - \theta_*)$

To begin, we will show that $\sqrt{n}(\hat{\theta}_n - \theta_*)$ behaves as a Gaussian random variable as $n \to \infty$. Recall that the empirical NCE loss is given by eq. (2):

$$L^n(\theta) = \frac{1}{n}\sum_{i=1}^n -\frac{1}{2}\ln\frac{p_\theta(x_i)}{p_\theta(x_i) + q(x_i)} + \frac{1}{n}\sum_{i=1}^n -\frac{1}{2}\ln\frac{q(y_i)}{p_\theta(y_i) + q(y_i)},$$

where $x_i \sim P_*$ and $y_i \sim Q$ are i.i.d. Let $\hat{\theta}_n$ be the optimizer for $L^n$. Then, by the Taylor expansion of $\nabla_\theta L^n$ around $\theta_*$, we have

$$\sqrt{n}\left(\hat{\theta}_n - \theta_*\right) = -\nabla_\theta^2 L^n(\theta_*)^{-1} \cdot \sqrt{n}\nabla_\theta L^n(\theta_*) - \sqrt{n}\cdot O\left(\left\|\hat{\theta}_n - \theta_*\right\|^2\right) \tag{13}$$

by Gutmann & Hyvärinen (2012), who also show in their Theorem 2 that $\hat{\theta}_n$ is a consistent estimator of $\theta_*$; hence, as $n \to \infty$, $\left\|\hat{\theta}_n - \theta_*\right\|^2 \to 0$. Gutmann & Hyvärinen (2012, Lemma 12) also assert[1] that the Hessian of the empirical NCE loss (eq. (2)) at $\theta_*$ converges in probability to the Hessian of the true NCE loss (definition 1) at $\theta_*$, i.e., $\nabla_\theta^2 L^n(\theta_*)^{-1} \xrightarrow{P} \nabla_\theta^2 L(\theta_*)^{-1}$. On the other hand, by the Central Limit Theorem, $\sqrt{n}\nabla_\theta L^n(\theta_*)$ converges to a Gaussian with mean $\mathbb{E}[\sqrt{n}\nabla_\theta L^n(\theta_*)] = \sqrt{n}\nabla_\theta L(\theta^*) = 0$, and covariance $\text{Var}[\sqrt{n}\nabla_\theta L^n(\theta_*)]$. With these considerations, we conclude that the random variable $\sqrt{n}(\hat{\theta}_n - \theta_*)$ in eq. (13) is asymptotically a Gaussian with mean 0 and covariance $\Sigma = \nabla_\theta^2 L(\theta_*)^{-1}\text{Var}[\sqrt{n}\nabla_\theta L^n(\theta_*)]\nabla_\theta^2 L(\theta_*)^{-1}$, as defined in eq. (12).

Next, we introduce some quantities which will be useful in the subsequent calculations. As we already have a handle on the spectrum of $\nabla_\theta^2 L(\theta_*)$ from Theorem 3, the main object of our focus in eq. (12) is the term $\text{Var}[\sqrt{n}\nabla_\theta L^n(\theta_*)]$. In particular, since we are concerned with the directional variance of $\Sigma$, we will reason about $\text{Var}\big[v^\top \cdot \sqrt{n}\nabla_\theta L^n(\theta_*)\big]$ for a fixed vector of ones, i.e., $v = 1^{d+1}$. This vector has the property that for all $x$, $v^\top T(x) \geq 1$, as all non-constant coordinates of $T$ are non-negative, and the remaining coordinate is 1. Note that

$$\nabla_\theta L^n(\theta_*) = -\frac{1}{2n}\sum_{i=1}^n \frac{q(x_i)T(x_i)}{p_*(x_i) + q(x_i)} + \frac{1}{2n}\sum_{i=1}^n \frac{p_*(y_i)T(y_i)}{p_*(y_i) + q(y_i)}$$

where $x_i \sim P_*$ and $y_i \sim Q$. Writing out the variance term explicitly, we have

$$\text{Var}\big[v^\top\sqrt{n}\nabla_\theta L^n(\theta_*)\big] = n \cdot \frac{1}{4n}\text{Var}_{x\sim p_*}\left[\frac{q(x)\cdot v^\top T(x)}{p_*(x) + q(x)}\right] + n\cdot\frac{1}{4n}\text{Var}_{y\sim q}\left[\frac{p_*(y)\cdot v^\top T(y)}{p_*(y) + q(y)}\right]$$

(using linearity and independence)

$$= \frac{1}{4}\text{Var}_{x\sim p_*}\underbrace{\left[\frac{q(x)\cdot v^\top T(x)}{p_*(x) + q(x)}\right]}_{A(x)} + \frac{1}{4}\text{Var}_{y\sim q}\underbrace{\left[\frac{p_*(y)\cdot v^\top T(y)}{p_*(y) + q(y)}\right]}_{B(y)}. \tag{14}$$

Define $A(x) = \frac{q(x)\cdot v^\top T(x)}{p_*(x) + q(x)} = \frac{R_1(x)}{1 + R_1(x)}v^\top T(x)$ where $R_1(x) = \frac{q(x)}{p_*(x)}$ and $B(y) = \frac{p_*(y)\cdot v^\top T(y)}{p_*(y) + q(y)} = \frac{R_2(y)}{1 + R_2(y)}v^\top T(y)$ where $R_2(y) = \frac{p_*(y)}{q(y)}$. To show that $\text{Var}_{x\sim p_*}[A(x)]$ and $\text{Var}_{y\sim q}[B(y)]$ are large, we will need anti-concentration bounds on $R_1(x)$ and $R_2(y)$.

---

[1] Translating notation: $T_d = n$, $J_{T_d}(\theta) = -2L^n(\theta)$ and setting $\nu = 1$ gives $\mathcal{I}_\nu = 2\nabla^2 L(\theta_*)$ as in eq. (6).

## 4.2 ANTI-CONCENTRATION OF $R_1(x), R_2(y)$

Next, we show that $R_1$ and $R_2$ satisfy (quantitative) anti-concentration. We show this by a relatively straightforward application of the Berry-Esseen Theorem, and the proof is given in Appendix B. Precisely, we show:

**Lemma 8.** *Let $d > 0$ be sufficiently large. Let $p = \hat{p}^d$ and $q = \hat{q}^d$ be any product distributions, and define $R(x) = \frac{q(x)}{p(x)}$. Suppose we have the following third moment bound: $\mathbb{E}_{x \sim \hat{p}}\left[\left(\log \frac{\hat{q}}{\hat{p}}\right)^3\right] < \infty$. Then, for any $\epsilon$, there exist constants $\alpha = \alpha(\hat{p}, \hat{q}, \epsilon)$, $\mu = \mu(\hat{p}, \hat{q}, \epsilon) < 0$ such that*

$$\mathbb{P}_{x \sim p}\left[R(x) \leq \exp\left(\mu d - \alpha\sqrt{d}\right)\right] \geq \frac{1}{2} - \epsilon \text{ and } \mathbb{P}_{x \sim p}\left[R(x) \geq \exp\left(\mu d + \alpha\sqrt{d}\right)\right] \geq \frac{1}{2} - \epsilon.$$

Instantiating Lemma 8 for the pair $(p_*, q)$ gives us the anti-concentration result for $R_1$, while instantiating it for the reversed pair $(q, p_*)$ gives us the anti-concentration result for $R_2$. We can verify that the third moment condition holds in both instantiations, since in both the cases, $\log(\hat{q}/\hat{p})$ is a polynomial. Crucially, we will also utilize the fact that the constant $\mu$ is negative (as it equals $-\mathrm{KL}(\hat{p}||\hat{q})$). We are now ready to bound the variance of $A(x)$ and $B(y)$.

## 4.3 BOUNDING THE VARIANCE OF $A(x), B(y)$

Recall that $A(x) = \frac{R_1(x) \cdot v^\top T(x)}{1 + R_1(x)}$ and $B(y) = \frac{R_2(y) \cdot v^\top T(y)}{1 + R_2(y)}$. Let $\mu, \alpha$ be the constants given by Lemma 8 for $p_*, q, \epsilon$. Further, let $L_1 = \exp\left(\mu d - \alpha\sqrt{d}\right)$ and $L_2 = \exp\left(\mu d + \alpha\sqrt{d}\right)$. Since the mapping $x \mapsto \frac{x}{1+x}$ is monotonically increasing in $x$,

$$\mathbb{P}_{x \sim p_*}[R_1(x) \leq L_1] = \mathbb{P}_{x \sim p_*}\left[\frac{R_1(x)}{1 + R_1(x)} \leq \frac{L_1}{1 + L_1}\right] \geq \frac{1}{2} - \epsilon \tag{15}$$

$$\mathbb{P}_{x \sim p_*}[R_1(x) \geq L_2] = \mathbb{P}_{x \sim p_*}\left[\frac{R_1(x)}{1 + R_1(x)} \geq \frac{L_2}{1 + L_2}\right] \geq \frac{1}{2} - \epsilon. \tag{16}$$

Let $T_{\mathrm{up}}$ be such that

$$\mathbb{P}_{x \sim p_*}\left[\|T(x)\| \leq T_{\mathrm{up}}\right] \geq \frac{7}{8} \quad \text{and} \quad \mathbb{P}_{x \sim q}\left[\|T(x)\| \leq T_{\mathrm{up}}\right] \geq \frac{7}{8}. \tag{17}$$

In Appendix D, we show that some $T_{\mathrm{up}} = O(\sigma^2\sqrt{d})$ suffices for this to hold. Then, from eq. (15), we have

$$\mathbb{P}_{x \sim p_*}\left[\frac{R_1(x)}{1 + R_1(x)} \leq \frac{L_1}{1 + L_1}\right] \geq \frac{1}{2} - \epsilon$$

$$\implies \mathbb{P}_{x \sim p_*}\left[\frac{R_1(x) \cdot v^\top T(x)}{1 + R_1(x)} \leq \frac{L_1\sqrt{d+1}\|T(x)\|}{1 + L_1}\right] \geq \frac{1}{2} - \epsilon \qquad \text{(Cauchy-Schwarz)}$$

$$\implies \mathbb{P}_{x \sim p_*}\left[\left(\frac{R_1(x) \cdot v^\top T(x)}{1 + R_1(x)} \leq \frac{L_1\sqrt{d+1}\|T(x)\|}{1 + Li_1}\right) \wedge \left(\|T(x)\| \leq T_{\mathrm{up}}\right)\right] \geq \frac{3}{8} - \epsilon$$
$$\text{(union bound with eq. (17))}$$

$$\implies \mathbb{P}_{x \sim p_*}\left[\frac{R_1(x) v^\top T(x)}{1 + R_1(x)} \leq \frac{\sqrt{d+1} L_1 T_{\mathrm{up}}}{1 + L_1}\right] \geq \frac{3}{8} - \epsilon$$

$$\implies \mathbb{P}_{x \sim p_*}\left[A(x) \leq \frac{\sqrt{d+1} L_1 T_{\mathrm{up}}}{1 + L_1}\right] \geq \frac{1}{4},$$

for $\epsilon \leq \frac{1}{8}$. On the other hand, recall also that $v$ satisfies $v^\top T(x) \geq 1$ for all $x$. Therefore, we have

$$\mathbb{P}_{x \sim p_*}\left[\frac{R_1(x)}{1 + R_1(x)} \geq \frac{L_2}{1 + L_2}\right] \geq \frac{1}{2} - \epsilon$$

$$\implies \mathbb{P}_{x \sim p_*}\left[\frac{R_1(x) \cdot v^\top T(x)}{1 + R_1(x)} \geq \frac{L_2}{1 + L_2}\right] \geq \frac{1}{2} - \epsilon \implies \mathbb{P}_{x \sim p_*}\left[A(x) \geq \frac{L_2}{1 + L_2}\right] \geq \frac{1}{4}.$$

Now, consider the event $A_1 = \left\{ A(x) \in \left[ \frac{1}{2}\mathbb{E}_{x \sim p_*}[A(x)], \frac{3}{2}\mathbb{E}_{x \sim p_*}[A(x)] \right] \right\}$. If this event were to intersect both the events $A_2 = \left\{ A(x) \leq \frac{\sqrt{d+1}L_1 T_{\mathrm{up}}}{1+L_1} \right\}$ and $A_3 = \left\{ A(x) \geq \frac{L_2}{1+L_2} \right\}$, then we would have

$$\frac{1}{2}\mathbb{E}_{x \sim p_*}[A(x)] \leq \frac{\sqrt{d+1}L_1 T_{\mathrm{up}}}{1+L_1} \quad \text{and} \quad \frac{3}{2}\mathbb{E}_{x \sim p_*}[A(x)] \geq \frac{L_2}{1+L_2}$$

$$\implies \quad \frac{L_2}{L_1} \cdot \frac{1}{T_{\mathrm{up}}\sqrt{d+1}} \cdot \frac{L_1+1}{L_2+1} \leq 3.$$

We will show that this cannot be the case. Recall that $\mu < 0$, which means that $L_2 = \exp(\mu d + \alpha\sqrt{d}) < 1$ for sufficiently large $d$. This means that for sufficiently large $d$ we have:

$$\exp(\mu d + \alpha\sqrt{d}) < 1$$
$$\implies \quad \exp(\mu d + \alpha\sqrt{d}) - 2\exp(\mu d - \alpha\sqrt{d}) < 1$$
$$\implies \quad 1 + \exp(\mu d + \alpha\sqrt{d}) < 2 + 2\exp(\mu d - \alpha\sqrt{d})$$
$$\implies \quad \frac{1 + \exp(\mu d - \alpha\sqrt{d})}{1 + \exp(\mu d + \alpha\sqrt{d})} > \frac{1}{2}$$
$$\implies \quad \frac{L_1+1}{L_2+1} > \frac{1}{2}.$$

Further, since $\frac{L_2}{L_1} = \exp(2\alpha\sqrt{d})$ and $T_{\mathrm{up}} = O(\sigma^2\sqrt{d})$, we get that

$$\frac{L_2}{L_1} \cdot \frac{1}{T_{\mathrm{up}}\sqrt{d+1}} \cdot \frac{L_1+1}{L_2+1} > \frac{\exp(2\alpha\sqrt{d})}{O(\sigma^2 d)} \cdot \frac{1}{2} > 3,$$

where the last inequality follows for large enough $d$ since the numerator grows faster than the denominator. Hence for large enough $d$, $A_1$ cannot intersect both $A_2$ and $A_3$. If the event $A_1$ is disjoint from $A_2$, then

$$\mathbb{P}_{x \sim p_*}[A_1 \cup A_2] = \mathbb{P}_{x \sim p_*}[A_1] + \mathbb{P}_{x \sim p_*}[A_2] \leq 1$$
$$\implies \quad \mathbb{P}_{x \sim p_*}[A_1] \leq 1 - \mathbb{P}_{x \sim p_*}[A_2]$$
$$\implies \quad \mathbb{P}_{x \sim p_*}\left[ A(x) \in \left[ \frac{1}{2}\mathbb{E}_{x \sim p_*}[A(x)], \frac{3}{2}\mathbb{E}_{x \sim p_*}[A(x)] \right] \right] \leq \frac{3}{4}$$
$$\implies \quad \mathbb{P}_{x \sim p_*}\left[ |A - \mathbb{E}_{p_*} A| \geq \frac{1}{2}\mathbb{E}_{p_*} A \right] \geq \frac{1}{4}.$$

This finally lower-bounds the variance of $A$ as

$$\mathrm{Var}_{p_*}[A] = \mathbb{E}\left[ (A - \mathbb{E}_{p_*} A)^2 \right] \geq \frac{1}{4}(\mathbb{E}_{p_*} A)^2 \cdot \mathbb{P}\left[ (A - \mathbb{E}_{p_*} A)^2 \geq \frac{1}{4}(\mathbb{E}_{p_*} A)^2 \right] \geq \frac{1}{16}(\mathbb{E}_{p_*} A)^2.$$

and thus $\mathbb{E}_{p_*}(A^2) - (\mathbb{E}_{p_*} A)^2 = \mathrm{Var}_{p_*}[A] \geq \frac{1}{16}(\mathbb{E}_{p_*} A)^2$, so that $(\mathbb{E}_{p_*} A)^2 \leq \frac{16}{17}\mathbb{E}_{p_*}(A^2)$.

Altogether, we get $\mathrm{Var}_{p_*}[A] \geq \frac{1}{17}\mathbb{E}_{p_*}(A^2)$. An analogous argument in the case when $A_1$ is disjoint with $A_3$ yields the same bound on the variance. Using an identical argument for $R_2$ and $B$, we get that for large enough $d$, $\mathrm{Var}_q[B] \geq \frac{1}{17}\mathbb{E}_q(B^2)$.

## 4.4 PUTTING THINGS TOGETHER

Putting together the lower bounds $\mathrm{Var}_{p_*}[A] \geq \frac{1}{17}\mathbb{E}_{p_*}(A^2)$ and $\mathrm{Var}_q[B] \geq \frac{1}{17}\mathbb{E}_q(B^2)$ we showed in the previous subsection, and recalling eq. (14), we get

$$\mathrm{Var}\left[ v^\top \cdot \sqrt{n}\nabla_\theta L^n(\theta_*) \right] = \frac{1}{4}\mathrm{Var}_{p_*}[A] + \frac{1}{4}\mathrm{Var}_{p_*}[B] \geq \frac{1}{68}\left( \mathbb{E}_{p_*}\left[ A^2 \right] + \mathbb{E}_q\left[ B^2 \right] \right)$$
$$= \frac{1}{68}\left( \int_x \left( \frac{q(x)^2 p_*(x) + q(x)p_*(x)^2}{(p_*(x) + q(x))^2} \right) v^\top T(x) T(x)^\top v\, dx \right)$$
$$= \frac{1}{68}v^\top \cdot \int_x \frac{p_*(x)q(x)}{p_*(x) + q(x)} T(x) T(x)^\top dx \cdot v = \frac{1}{34}v^\top \nabla_\theta^2 L(\theta_*) v$$

$$\text{(from eq. (6)).}$$

Finally, since $\nabla_\theta^2 L(\theta_*)$ is invertible as claimed earlier (Lemma 10, Appendix C), let $w$ be such that $v = \nabla_\theta^2 L(\theta_*)^{-1} w$. Then, recalling the expression for $\Sigma$ in eq. (12), we can conclude that

$$w^\top \Sigma w = v^\top \mathrm{Var}\big[\sqrt{n}\nabla_\theta L^n(\theta_*)\big]v = \mathrm{Var}\big[v^\top \cdot \sqrt{n}\nabla_\theta L^n(\theta_*)\big]$$
$$\geq \frac{1}{34}v^\top \nabla_\theta^2 L(\theta_*)v = \frac{1}{34}w^\top \nabla_\theta^2 L(\theta_*)^{-1}w, \qquad (18)$$

which gives us the desired bound on the MSE, namely

$$\mathbb{E}\bigg[\Big\|\hat{\theta}_n - \theta_*\Big\|_2^2\bigg] \geq \frac{1}{n}\mathrm{Tr}(\Sigma) \geq \frac{1}{n}\sup_z \frac{z^\top \Sigma z}{\|z\|^2}$$
$$\geq \frac{1}{n}\frac{w^\top \Sigma w}{\|w\|^2} \geq \frac{1}{34n}\frac{w^\top \nabla_\theta^2 L(\theta_*)^{-1}w}{\|w\|^2} \geq \frac{1}{34n}\inf_z \frac{z^\top \nabla_\theta^2 L(\theta_*)^{-1}z}{\|z\|^2} \geq \frac{\exp(\Omega(d))}{n},$$

where the last inequality follows from Theorem 3 and the fact that $\lambda_{\max}(\nabla_\theta^2 L(\theta_*))^{-1} = \lambda_{\min}(\nabla_\theta^2 L(\theta_*)^{-1})$. This concludes the proof of Theorem 4.

## 5 SIMULATIONS

We also verify our results with simulations. Precisely, we study the MSE for the empirical NCE loss as a function of the ambient dimension, and recover the dependence from Theorem 4. For dimension $d \in \{70, 72, \ldots, 120\}$, we generate $n = 500$ samples from the distribution $P_*$ we construct in the theorem. We generate an equal number of samples from the noise distribution $Q = \mathcal{N}(0, I_d)$, and run gradient descent to minimize the empirical NCE loss to obtain $\hat{\theta}_n$. Since we explicitly know what $\theta_*$ is, we can compute the squared error $\|\hat{\theta}_n - \theta_*\|^2$. We run 100 trials of this, where we obtain fresh samples each time from $P_*$ and $Q$, and average the squared errors over the trials to obtain an estimate of the MSE.

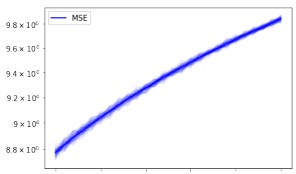

Figure 1: Log MSE versus Dimension—Theorem 4 suggests this plot should be linear, as is observed.

Figure 1 shows the plot of $\log$ MSE versus dimension - we can see that the graph is nearly linear. This corroborates the bound in Theorem 4, which tells us that as $n \to \infty$, the MSE scales exponentially with $d$. This behavior is robust even when the proportion of noise samples to true data samples is changed to 70:30 (though our theory only addresses the 50:50 case). Finally, we note that optimizing the empirical NCE loss becomes numerically unstable with increasing $d$ (due to very large ratios in the loss), which is why we used comparatively moderate values of $d$.

## 6 CONCLUSION

Despite significant interest in alternatives to maximum likelihood—for example NCE (considered in this paper), score matching, etc.—there is little understanding of what there is to "sacrifice" with these losses, either algorithmically or statistically. In this paper, we provided formal lower bounds on the asymptotic sample complexity of NCE, when using a common choice for the noise distribution $Q$, a Gaussian with matching mean and covariance. Thus, it is likely that even for moderately complex distributions in practice, more involved techniques like Gao et al. (2020); Rhodes et al. (2020) will have to be used, in which one learns a noise distribution $Q$ simultaneously with the NCE minimization or "anneals" the NCE objective. There is very little theoretical understanding of such techniques, and this seems like a very fruitful direction for future work.

## ACKNOWLEDGEMENTS

Andrej is supported in part by NSF award IIS-2211907.

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

## A  BOUNDING THE MATRIX INTEGRAL IN EQUATION 6

We prove a variant of the Cauchy-Schwarz inequality that gives us a handle on norms of matrix integrals.

**Lemma 9.** *Let $f : \mathbb{R}^d \to \mathbb{R}$ and $A : \mathbb{R}^d \to \mathbb{R}^n$ be integrable functions, with $M = \int_x f(x)A(x)dx$. Then we have*

$$\|M\|_2^2 = \left\|\int_x f(x)A(x)ds\right\|_2^2 \leq \left(\int_x |f(x)|^2 dx\right)\left(\int_x \|A(x)\|_2^2 dx\right). \tag{19}$$

*Similarly, if $A : \mathbb{R}^d \to \mathbb{R}^{n \times n}$ is a matrix valued function then*

$$\|M\|_F^2 = \left\|\int_x f(x)A(x)\right\|_F^2 \leq \left(\int_x |f(x)|^2 dx\right)\left(\int_x \|A(x)\|_F^2 dx\right). \tag{20}$$

*Proof.* The proof follows from the Cauchy-Schwarz inequality. Since we integrate component-wise, for eq. (19) we have that

$$M_i^2 = \left(\int_x f(x)A(x)_i dx\right)^2 \leq \left(\int_x f(x)^2 dx\right)\left(\int_x A(x)_i^2 dx\right).$$

Summing over $i$, we get the result. The matrix variant eq. (20) follows by looking at the matrix $M$ as a vector in $\mathbb{R}^{n^2}$. $\qquad \square$

## B  PROOF OF LEMMA 8

We restate the lemma for convenience:

**Lemma 8.** *Let $d > 0$ be sufficiently large. Let $p = \hat{p}^d$ and $q = \hat{q}^d$ be any product distributions, and define $R(x) = \frac{q(x)}{p(x)}$. Suppose we have the following third moment bound: $\mathbb{E}_{x \sim \hat{p}}\left[\left(\log \frac{\hat{q}}{\hat{p}}\right)^3\right] < \infty$. Then, for any $\epsilon$, there exist constants $\alpha = \alpha(\hat{p}, \hat{q}, \epsilon)$, $\mu = \mu(\hat{p}, \hat{q}, \epsilon) < 0$ such that*

$$\mathbb{P}_{x \sim p}\left[R(x) \leq \exp\left(\mu d - \alpha \sqrt{d}\right)\right] \geq \frac{1}{2} - \epsilon \text{ and } \mathbb{P}_{x \sim p}\left[R(x) \geq \exp\left(\mu d + \alpha \sqrt{d}\right)\right] \geq \frac{1}{2} - \epsilon.$$

*Proof.* We will analyze the behaviour of $R(x)$ using the Berry-Esseen theorem. Given that $p_* = \hat{p}^d$ and $q = \hat{q}^d$ are product distributions, let $r(x)$ be the random variable defined by $r(x) = \frac{\hat{q}(x)}{\hat{p}(x)}$, $x \sim \hat{p}$. Let $y_i(x) = \log r(x)$ for $1 \leq i \leq d$ be $d$ independent copies of the random variable $r(x)$. Let $\mathbb{E}[y_i] = \mu_r$, $\mathbb{E}\left[\|y_i - \mu_r\|^2\right] = \sigma_r^2$ and $\mathbb{E}\left[\|y_i - \mu_r\|^3\right] = \gamma_r$, all of which are well defined by the hypothesis of the lemma. Let $Y = \sum_{i=1}^d y_i$, and $Z$ be the standard Gaussian in $\mathbb{R}$. Then, by the Berry-Esseen Theorem (Durrett, 2019, Theorem 3.4.17),

$$\mathbb{P}\left[\frac{Y - \mu_r d}{\sigma_r \sqrt{d}} \leq -c\right] \geq \mathbb{P}[Z \leq -c] - \frac{C_{\text{BE}} \cdot \gamma_r}{\sigma_r^3 \sqrt{d}},$$

where $C_{\text{BE}} < 1$ (van Beek, 1972) is an absolute constant. We can now choose $c = c(\epsilon)$ such that $\mathbb{P}[Z \leq c] \geq \frac{1-\epsilon}{2}$. Further, we can choose $d$ large enough so that $\frac{C_{\text{BE}} \cdot \gamma}{\sigma^3 \sqrt{d}} \leq \frac{\epsilon}{2}$. Then for $\mu = \mu_r$ and $\alpha = c\sigma_r$, we have

$$\mathbb{P}_{x \sim p}\left[R(x) \leq \exp\left(\mu d - \alpha \sqrt{d}\right)\right] \geq \frac{1}{2} - \epsilon.$$

Since $Z$ is symmetric around 0, Berry-Esseen gives us the other inequality for the same choice of $\mu$ and $\alpha$,

$$\mathbb{P}\left[\frac{Y - \mu_r d}{\sigma_r \sqrt{d}} \geq c\right] \geq \mathbb{P}[Z \geq c] - \frac{C_{\text{BE}} \cdot \gamma_r}{\sigma_r^3 \sqrt{d}} \geq \frac{1}{2} - \epsilon.$$

Note that the constants $\mu$ and $\alpha$ are independent of $d$. Further, note that $\mu = \mu_r = -\text{KL}(\hat{p}||\hat{q}) < 0$. $\qquad \square$

## C  INVERTIBILITY OF THE HESSIAN

We prove that the Hessian of NCE loss for the exponential family given by $T(x) = (x_1^4, \ldots, x_d^4, 1)$ is invertible. In particular, we have the following lemma:

**Lemma 10.** *Let $Q = \mathcal{N}(0, I_d)$ be the standard Gaussian in $\mathbb{R}^d$. Let $\hat{P}$ be the log concave distribution defined in definition 6. Let $P = \hat{P}^d$. Let $q$ and $p$ denote the density functions of $Q$ and $P$ respectively. Observe that $P$ is in the exponential family given by $T(x) = (x_1^4, \ldots, x_d^4, 1)$, and equals $P_{\theta_*}$ for some $\theta_*$. Then the hessian of the NCE loss with respect to distribution $P$ and noise $Q$ given by*

$$H = \nabla_\theta^2 L(\theta_*) = \frac{1}{2} \int_x \frac{p_* q}{p_* + q} T(x) T(x)^\top$$

*is invertible.*

*Proof.* For any subset $A \subseteq \mathbb{R}^d$, define

$$H_A = \frac{1}{2} \int_{x \in A} \frac{p_* q}{p_* + q} T(x) T(x)^\top.$$

Observe that the density functions $p_*$ and $q$ of $P_*$ and $Q$ respectively are strictly positive over all of $\mathbb{R}^d$. Therefore, for any subset $A \subseteq \mathbb{R}^d$ and any $v \in \mathbb{R}^{d+1}$, we have

$$v^\top H v \geq \frac{1}{2} \int_{x \in A} \frac{p_* q}{p_* + q} v^\top T(x) T(x)^\top v = v^\top H_A v.$$

Given a vector $v \in \mathbb{R}^{d+1}$, we will pick $A$ such that $\left| T(x)^\top v \right| > 0$ for all $x \in A$. Note that the set $\mathcal{B} = \{e_1 + e_{d+1}, \ldots, e_d + e_{d+1}, e_{d+1}\}$ is a basis. Therefore, if $b^\top v = 0$ for all $b \in \mathcal{B}$, then $v = 0$. Hence, there exists some $x \in \{e_1, \ldots, e_d\}$ such that $\left| T(x)^\top v \right| > 0$. Since $x \mapsto T(x)^\top v$ is a continuous function, we can find an open set $A$ around $x$ such that

$$\left| T(y)^\top v \right| > 0, \qquad \forall y \in A.$$

It follows that

$$v^\top H_A v = \frac{1}{2} \int_{x \in A} \frac{p_* q}{p_* + q} v^\top T(x) T(x)^\top v = \frac{1}{2} \int_{x \in A} \frac{p_* q}{p_* + q} \left| T(x)^\top v \right|^2 > 0.$$

Let $B = \mathbb{R}^d \setminus A$. Since $v^\top H_A v > 0$ and $v^\top H_B v \geq 0$, we have that $v^\top H v > 0$. Since this holds for any arbitrary non-zero vector $v$, the matrix $H$ must be full rank. Since $H$ is an integral of PSD matrices, it is a full rank PSD matrix and hence invertible. $\square$

## D  TAIL BOUNDS FOR EQUATION 17

We prove that some $T_{\mathrm{up}} = O(\sigma^2 \sqrt{d})$ suffices to obtain the bounds in eq. (17). Concretely, we prove tail bounds for $\|T(x)\|$ using tail bounds for $P_*$ and $Q$. We will use Lemma 1 from Laurent & Massart (2000) which proves a bound for $\chi^2$ distributions:

**Lemma** (Lemma 1, Laurent & Massart (2000)). *If $X$ is a $\chi^2$ random variable with $d$ degrees of freedom, then for any positive $t$,*

$$\mathbb{P}\left[ X - d \geq 2\sqrt{td} + 2t \right] \leq \exp(-t).$$

Then, for $x \sim Q$, $\|x\|^2$ is a $\chi^2$ random variable with $d$ degrees of freedom. Observe that for $t, d \geq 4$, we have $d + 2t + 2\sqrt{td} \leq 2td$. In particular, we have the weaker bound

$$\mathbb{P}_{x \sim Q}\left[ \|x\|^2 \geq 2dt^2 \right] \leq \exp(-t^2),$$

implying that

$$\mathbb{P}_{x \sim Q}\left[ \|x\| \geq t \right] \leq \exp\left( -\frac{t^2}{2d} \right).$$

Further, if $\|x\| \geq \sigma^2 \sqrt{d}$, $q(x) \geq p_*(x)$, implying that for $t \geq \sigma^2 \sqrt{d}$

$$\mathbb{P}_{x \sim P_*}\left[\|x\| \geq t\right] \leq \exp\left(-\frac{t^2}{2d}\right).$$

In particular, for any $\delta$ such that $\log(1/\delta) \geq \sigma^4$, we have

$$\mathbb{P}_{x \sim Q}\left[\|x\| \geq \sqrt{2d \log(1/\delta)}\right] \leq \delta \qquad \text{and} \qquad \mathbb{P}_{x \sim P_*}\left[\|x\| \geq \sqrt{2d \log(1/\delta)}\right] \leq \delta. \qquad (21)$$

