# OpenReview forum: "Pitfalls of Gaussians as a noise distribution in NCE"
_ICLR.cc/2023/Conference — ICLR 2023 poster_

### Official Review · Reviewer_mHes · 2022-10-23

**Confidence:** 4
**Correctness:** 4
**Technical Novelty And Significance:** 4
**Empirical Novelty And Significance:** 3
**Recommendation:** 8

**Clarity, Quality, Novelty And Reproducibility:**

The paper is clear, short, easy, and nice to read.

I verified virtually all of the math in it. It's nice and clean and short math.

As I mentioned in the last box, I don't know much about the novelty or significance of it.


---

I don't have anything else to write here, so I'll add my typo/small edits list here:

1. [Page 1, second to last paragraph] You mention (Liu et al 2021) as another theoretical work, and mention them again later in the paper, discussing the theoretical hardness of NCE. You should explain how your result differs from theirs.
1. [Page 2, just after theorem 3] Say "We remark that our above problematic" instead of "By way of remarks, note that the above example of a problematic"
1. [Page 2, last paragraph] "seriously" sounds weird here. Maybe use "concerningly"?
1. [Page 3, first paragraph, second sentence] Break open this run-on sentence into at least 3 sentences. It's kinda confusing when all stitched into a single sentence with a bunch of commas.
1. [Page 3, just before equation 3] "forms" not "form"
1. [Page 3, just after equation 7] "matrix version" not "vector version"
1. [Page 3, just before section 3.2] Add a short paragraph here, looking at these two terms. The is small when P is very very very different from Q, so the less representative the noise distribution is of P, the smaller the Hessian is. The second term is Fisher-matrix-like, which is known to relate to the smoothness of optima, so it's not surprising it appears. But the TV is going to be so close to one, that this Fisher term will be irrelevant.
    - Also, I don't think this exactly a norm on the Fisher information matrix? It's definitely similar, but I don't see a way to bound the Fisher matrix by this term?
1. [Page 4, start of proof of lemma 5] Rewrite the statement of lemma 5 to just use this language. Something like "given a univariate distribution that's log-concave with mean zero and unit variance and has Bhattacharya $\rho$ with the N(0,1) gaussian, we can take the d-dimensional product distribution, and that will have TV $1-\rho^d$ with the isotropic gaussian". No need for this "there exists" language.
1. [Page 4, last line of lemma 5] Consider citing that the product of log-concaves is log-concave?
1. [Page 5, lemma 7] Is this the fisher information matrix? I'm still not convinced.
1. [Page 5, last paragraph] The last two sentences are hard for me to understand. I don't really know what "directional variance" is. You maybe just pull the analysis about $v$ and $w$ from the bottom of page 8 to be here, so that you can just concretely talk about how $w^\intercal \Sigma w = v^\intercal Var[...] v$ where $v$ is the all-ones vector?
1. [Page 6, Section 4.1] where do you actually use the fact that $\sqrt{n} (\hat\theta - \theta^*)$ is asymptotically Gaussian? Does it matter?
1. [Page 6, Variance equation in the middle] Consider adding a page of appendix showing this variance equation. You didn't write down the empirical NCE loss's gradient even, so the variance working out this way is at best a chore to verify by hand. This is the only line in the body of the paper that I didn't verify.
1. [Page 8, second-to-last line of math] Missing a start underneath $\theta$ in the variance.
1. [Page 9, first line of math] The first inequality should be an equality
1. [Page 9, experiments] The experiments have a good base but could easily be just a lil more satisfying. If you run 100 trials, then can you plot the 25th and 75th quantiles around the sample MSE. Also, could you plot a second line showing how 70:30 empirically performed. You said it was similar, but I'd like to see how similar it is. Lastly, the log-axis would read a bit better if instead of computing the logarithm yourself, you used a setting in the plotting software to make the y-axis show the true values just on a logarithmic y-axis. Ya know, like a log-linear plot. [Here's how to do it it matplotlib, though I don't know what software you're using](https://stackoverflow.com/questions/773814/plot-logarithmic-axes).
1. [Page 9, experiments] I'd say that it's "close to linear" instead of "linear". There's a pretty visible but very mild curvature. It's a real toss up, so do what you want.

**Strength And Weaknesses:**

_Note that I'm not an expert on NCE. I understand the math in the paper, and I'm good with statistical and machine learning lower bounds, but I can't really speak to how this paper fits into the broader NCE literature._

The paper is short and compelling. The math is short. The language is clear. The claim is somewhat interesting, though I suspect the authors may be underselling the underlying intuition that implies this exponential dependence. I'll elaborate on these points below.

First, the paper is short and the math is clean. I read almost the entire thing, and verified virtually every line of the math. I've got a couple minor questions, but more than almost any other theory paper, I can really claim this paper is mathematically correct.

Second, the paper is well written. The language is short. It doesn't waste your time. It quickly gets to the core ideas, has a nice logical flow, and the authors give nice intuitions preceding each step of the proof, so you never loose the big picture between the equations.

The claims and proofs feels significant and novel, though this is hard for me to judge as an outsider to the NCE literature.


That said, the first major claim of the paper (what I referred to as the "intermediate result" in my summary), is somewhat hard to interpret. First, keep in mind two facts:
1. NCE means minimizing a specific loss function (the _empirical NCE loss_; equation (2) on page 2 of the paper)
2. NCE is not tied to a specific optimization algorithm.

This first major claim says that the _population_ NCE loss function, which is impossible to compute since it requires knowing the distribution we're trying to approximate, is very flat near it's optimum. Near the bottom of page 2 of the paper, the authors claim this suffices to show that typical first-order methods would take exponentially long time to converge unless they make a mitigating factor like normalizing their gradients. Putting aside the fact that it is it pretty easy to normalize a gradient, the authors don't even prove that the empirical loss is flat near the optimum. So, it's not clear that this claim carries much weight on its own as an independent contribution.

That said, this is an important intermediate result for the final claim of the paper, which uses an anti-concentration argument to say that the mean squared error of NCE is bounded by the flatness of the _population_ NCE loss function. So it's important for the final result about NCE sample complexity, but I think those claims should be either better explained or toned down or just removed.

The final exponential sample complexity ends up relying on the exponential flatness (exponentially small spectral norm of the Hessian) of the population loss function. So, if we want to learn a broader lesson about when picking a bad noise distribution can make NCE require a huge sample complexity, we should look inside the proof of the exponentially small spectral norm, specifically at equation (8) on page 3:
$$\|\| \nabla_\theta^2 L(p_\theta) \|\| \leq \frac12 (1-TV(P,Q))^{1/2} \ \cdot \  (\textstyle{\int} \|\|T(x)T(x)\|\|_F^2 p(x) dx)^{1/2}$$
This right hand side has two terms:
1. A total variation that measure how far the true distribution is from the noise distribution. This term is exponentially close to zero.
2. Something akin to a norm on the Fisher Information Matrix, which is a known measure for the flatness of optima. This term is polynomially large.

This upper bound on the norm of the Hessian crucially assumes the true distribution is an exponential distribution, but that's all. The total variation is going to be exponentially small in the dimension any time both the noise and true distributions have $iid$ entries and when the noise distribution isn't exactly equal to the true distribution. The second term should be polynomially large in all reasonable circumstances. So, from my intuition at least, we expect to have this exponential flatness for a much much wider choice of noise distributions and true distributions. At least, we should carry this intuition around and have to argue why some other setting won't fall into this pitfall.

Basically, I suspect there's another interesting result that can be found by generalizing this paper's current results to a much larger family of distributions, showing that the exponential dependence on dimension is much more general than just the Gaussian noise distribution. I don't know how interesting that would be to the broader NCE community though.

**Summary Of The Paper:**

The paper analyzes Noise Contrastive Estimation (NCE): a popular algorithm for approximating distributions given iid samples from it, and whose effectiveness crucially depends on the user's choice of a "noise distribution" to compare the given samples to. Often, this noise distribution is chosen to just be an isotropic Gaussian.

This paper shows that even if the given samples are drawn iid from a extremely well behaved distribution, then NCE with the isotropic Gaussian as its noise distribution has a mean squared error that grows exponentially in the dimension of the data. In other words, for NCE to low mean squared error, the number of samples required must be exponential in the dimension of the data.

This "well behaved" distribution is a $d$-dimensional vector with $iid$ log-concave entries from an exponential distribution. It's basically a mild generalization of the isotropic gaussian, where the pdf depends on $e^{t^4}$ instead of $e^{t^2}$.

The paper emphasizes an intermediate result, showing that the NCE loss metric is exponentially flat near it's optimum. Critically, this result is for the population loss (which the algorithm doesn't optimize over), as opposed to the sample loss (which the population does optimize over). This distinction isn't made clear in the introduction or abstract.

**Summary Of The Review:**

I like this paper, and unless other reviewers have concerns around it's novelty and/or significance, I'd like to see it published.

Short an sweet. Interpretable results. I'm happy.

---

> ### Author Response · Authors · 2022-11-10
> **Response to Reviewer mHes**
>
> Thank you for your extremely thorough review and great summary of our paper! Thank you for also pointing out the typos and writing suggestions—we have implemented the changes in the updated draft (in brown). We address your concerns in the following:
>
> ---
>
> > *"Basically, I suspect there's another interesting result that can be found by generalizing this paper's current results to a much larger family of distributions, showing that the exponential dependence on dimension is much more general than just the Gaussian noise distribution"*
>
> You are correct, NCE might be statistically poorly behaved with other noise distributions as well!
> The reason we specifically focus on the Gaussian is that it is the canonical noise distribution chosen in practice, because it is: (1) easy to fit from the training data; (2) easy to sample from; (3) has an explicit expression for the density. Our proof only essentially uses the fact that $q$ is a product distribution, and that the coordinatewise marginals of $p^*$ and $q$ are at some constant total variation distance away from each other.
>
> The takeaway point of the paper isn’t that the Gaussian is a uniquely bad noise distribution: rather it is that distances between distributions tend to be amplified in high dimensions in a way that is detrimental to NCE, so that the noise distribution likely needs to be more carefully selected in high dimensions, in order for NCE to have good statistical efficiency.
>
> ---
>
> > *"Is this the Fisher Information Matrix"*
>
> You are right, technically the relevant factor in equation (8) is an upper bound on the Frobenius norm of the Fisher matrix. We updated the verbiage to match this in the relevant spots in the manuscript.
>
> ---
>
> We hope that this clarifies things, and we would be happy to answer any further questions that you have.

---

> > ### Comment · Reviewer_mHes · 2022-11-17
> > **Pedantic question about the Fisher info matrix**
> >
> > Hey there,
> >
> > **For the meta-reviewer**: I'm confidently holding my score of 8. This paper should be accepted.
> >
> > ---
> >
> > On the first part -- that sounds good. The traits you say sound realllly general, so it may be worth writing down a (formal or maybe even informal) very general theorem/proof that basically says "If you noise distribution is not suuuper close to the real data distribution, then you will have an exponentially hard time learning the real distribution". A statement that really talks about any data distribution and any noise distribution, and perhaps uses the Gaussian as a specific demonstrative example, since the Gaussian is so ubiquitous.
> >
> > That feels like a stronger story you should be writing about -- not only do Gaussians not suffice but _anything far from the real distribution_ does not suffice. Feel free to not write it out I guess, you're getting an 8 from me either way, but I'd recommend adding it to the text.
> >
> > ---
> >
> > Okay this is pedantic so legit feel free not to respond to this, but also I don't see how this is an upper bound on the frob norm (squared?) of the Fisher info matrix. Maybe I'm working from the wrong starting point, but this is what I'm seeing:
> > $$ \begin{align}
> > I_{i,j}(\theta)
> > &= \mathbb{E}[ \textstyle{(\frac d{d\theta_i} \log p_\theta(x)) \cdot (\frac d{d\theta_j} \log p_\theta(x))} \\\\
> > &= \mathbb{E}[ [T(x)]_i \cdot [T(x)]_j ] \\\\
> > &= \int [T(x)]_i \cdot [T(x)]_j p_\theta(x) dx \\\\
> > I(\theta) &= \int T(x) (T(x))^\intercal p_\theta(x) dx \\\\
> > \|\|I(\theta)\|\|_F &= \left\|\left\|\int T(x) (T(x))^\intercal p_\theta(x) dx \right\|\right\|_F \\\\
> > &\leq \int \|\|T(x) (T(x))^\intercal p_\theta(x)\|\|_F dx \\\\
> > \end{align} $$
> > But then I don't see how to get the square inside of the integral

---

> > > ### Author Response · Authors · 2022-11-18
> > > **Relation with the Fisher Info Matrix**
> > >
> > > I believe you get a bound with square root, that is
> > > $$
> > > \lVert I(\theta) \rVert^2_F
> > > = \sum\_{i,j} \left( \int T(x)_i T(x)_j p_\theta (x) dx \right)^2
> > > \le \sum\_{i,j} \left( \int T(x)_i^2 T(x)_j^2 p_\theta (x) dx \right)
> > > = \int \lVert T(x) T(x)^\intercal \rVert^2_F p_\theta (x) dx
> > > $$
> > > the second inequality is cauchy schwatz with measure $p_\theta (x)$ which implies that
> > > $$
> > > \lVert I(\theta) \rVert_F \le \left( \int \lVert T(x) T(x)^\intercal \rVert^2_F p_\theta (x) dx \right)^{\frac{1}{2}}
> > > $$
> > > the RHS is the second term in the paper

---

### Official Review · Reviewer_cCKs · 2022-10-24

**Confidence:** 4
**Correctness:** 4
**Technical Novelty And Significance:** 3
**Empirical Novelty And Significance:** Not applicable
**Recommendation:** 6

**Clarity, Quality, Novelty And Reproducibility:**

The paper is overall well-written and the results are written in a clean and
consice way.  I had no issues following the main claims of the paper.


**Strength And Weaknesses:**

Strengths

1. The NCE objective is a popular method and investigating its behavior under
different noise distribution is an important and well-motivated problem.  The
results of this work provide a concrete "counter-example" to a commonly used
noise distribution.

2. The paper is well-written and the proof details are easy to follow.

Weaknesses

1. Since the contribution of this work is theoretical, I think a minor weak
point of this work is that the proofs and technical results are not
particularly challenging.


Questions

I think for the parametric family constructed to make the NCE loss
exponentially flat, the maximum likelihood objective is strongly convex (for
some dimension-independent constant).  If this is true the authors could
perhaps add this small observation so that it is more clear that for the family
constructed the maximum objective would be efficiently optimizable.

Minor Typos

page 4, "the gamma function defined as $\Gamma(z) \int_0^\infty x^{z-1} e^{-x} d x$.



**Summary Of The Paper:**

This paper
considers estimating the parameters of an exponential family using the NCE
loss.  The NCE loss is a particularly popular method because it avoids
computing the partition function of the exponential family which is required by
other methods such as maximum likelihood.  A popular choice for the noise
distribution is to use a Gaussian that matches the mean and covariance of the
observations.  This work shows that there exist simple exponential families for
which such a choice makes the NCE exponentially flat around the optimal
parameter.  This fact also results in exponentially large sample complexity for
the NCE loss.  More precicely, the first result shows that for this exponential
family the $\ell_2$-norm of the Hessian of the NCE loss with Gaussian noise
distribution is at most $e^{-\Omega(d)}$, where $d$ is the dimension of the
observations.  They then use this result to show that the empirical optimizer
of the NCE loss needs exponentially many samples ($e^{\Omega(d)}$) in order to
converge to the true parameter (be in constant $\ell_2$ distance from the true
parameter).


**Summary Of The Review:**

This paper considers an interesting problem and provides a simple
"counter-example" for a common choice for the noise distribution in the NCE
loss.  Overall, I found the paper to be somewhat "light" in technical content
but I liked the fact that it provides a clean counter-example to popular method
and fills a gap in the literature on the NCE loss. Therefore, I am inclined
towards acception.

---

> ### Author Response · Authors · 2022-11-10
> **Response to Reviewer cCKs**
>
> Thank you for your positive review! We address your concerns in the following:
>
> ---
>
> > *“Since the contribution of this work is theoretical, I think a minor weak point of this work is that the proofs and technical results are not particularly challenging.”*
>
> We believe that one of the strengths of the paper is that the proofs are clean and intuitive—as pointed out by reviewer mHes. Moreover, we think this make the results more easy to adapt and generalize to other settings: for instance, since our proof only essentially uses the fact that $q$ is a product distribution, and that the coordinatewise marginals of $p^*$ and $q$ are at some constant total variation distance away from each other—one can show similar results for more general classes of noise distributions. We hope our techniques can also be used to prove limitations of other losses (e.g. the eNCE objective from Liu et al. [1], the generalized NCE loss from Uehara et al. [2]).
>
> [1]: Liu, B., Rosenfeld, E., Ravikumar, P. and Risteski, A. Analyzing and improving the optimization landscape of noise-contrastive estimation. ICLR 2022
> [2] Uehara, M., and Kanamori,T., and Takenouchi,T., and Matsuda, T. A Unified Statistically Efficient Estimation Framework for Unnormalized Models. AISTATS 2020.
>
> ---
>
> > *“I think for the parametric family constructed to make the NCE loss exponentially flat, the maximum likelihood objective is strongly convex (for some dimension-independent constant). If this is true the authors could perhaps add this small observation so that it is more clear that for the family constructed the maximum objective would be efficiently optimizable”*
>
> Correct, the MLE objective landscape for exponential families is in general convex. For general exponential families though, the partition function would be hard to calculate (it involves a high-dimensional integral), so gradient descent would be computationally intractable. In our special case, since the exponential family consists of product distributions, the partition function can in fact be efficiently estimated. This is however *not* a bug, rather a *feature* of our construction: it shows that the family of distributions we are considering is not “hard” to learn (statistically or computationally), yet NCE is statistically inefficient.
>
> ---
>
> We hope that this clarifies things, and we would be happy to answer any further questions that you have.

---

> ### Author Response · Authors · 2022-11-18
> **Response to Reviewer cCKs**
>
> We thank you again for your review !
>
> We hope that our response convinced you that the fact that our results are clean and intuitive is a feature, rather than a bug of our paper. Since the discussion period is coming to a close today, please let us know if there are any other questions we can answer.

---

### Official Review · Reviewer_9KJi · 2022-10-25

**Confidence:** 3
**Correctness:** 4
**Technical Novelty And Significance:** 2
**Empirical Novelty And Significance:** Not applicable
**Recommendation:** 8

**Clarity, Quality, Novelty And Reproducibility:**

### Quality and Clarity:

- The paper is well written and the proofs are rigorous and well-presented. The quality of the paper can be improved through a more detailed discussion of related work.

### Novelty:

- The paper largely builds upon central ideas in Liu et al. However, the theoretical results for statistical complexity and the construction of a multidimensional exponential distribution satisfying the hardness for NCE with Gaussian samples are novel and shed light on the limitations of gaussian noise for NCE.

**Strength And Weaknesses:**

### Strengths:

- The paper highlights the limitations of a common practical choice of gaussian distribution while using NCE.
- The paper is well-written.
- The proof technique of utilizing the tensorization of the Hellinger distance is interesting and non-trivial.
- The proofs are easy to follow.
- The results are supported by simulations.

### Weaknesses:

- The paper doesn't contain adequate discussion of related work. Including a related work section with a more detailed comparison against works such as Liu et al. [1] would improve the presentation of the paper.
- The poor algorithmic efficiency of optimization algorithms for NCE due to an exponentially flat Hessian of the loss was discussed in Liu et al. [1]. Therefore, the novelty of the paper appears to be limited.
- The relevance of the novel results for the use of eNCE and normalized gradient method in Liu et al. [1] is not discussed.
- A discussion of the relationship with related generative models such as Generative Adversarial Networks is absent.
- The effect of utilizing sample mean and covariance instead of the population mean, and covariance is not discussed.
- Possible approaches for mitigating the pitfalls of gaussian noise are not presented.

[1]: Liu, B., Rosenfeld, E., Ravikumar, P. and Risteski, A., 2021. Analyzing and improving the optimization landscape of noise-contrastive estimation. arXiv preprint arXiv:2110.11271.

**Summary Of The Paper:**

The paper analyzes the Noise Contrastive Estimation (NCE) algorithm under the choice of a gaussian sampling distribution with mean and covariance estimated from the data.
The paper shows that such a choice can lead to both poor statistical and computation efficiency by constructing an exponential distribution that leads to ill-conditioned Hessian with a spectral norm exponentially decaying in the ambient dimension. Such a choice also leads to exponentially bad statistical efficiency.

**Summary Of The Review:**

The paper is well-written and provides novel theoretical contributions to the analysis of the widely used algorithm NCE. The paper's novelty is however limited and the additional contributions of the paper compared to Liu et al. [1] should be discussed. The paper can also be improved through a discussion of suitable alternatives to gaussian noise and algorithms that are not significantly affected by the poor conditioning of the Hessian.

[1]: Liu, B., Rosenfeld, E., Ravikumar, P. and Risteski, A., 2021. Analyzing and improving the optimization landscape of noise-contrastive estimation. arXiv preprint arXiv:2110.11271.

____________

Post rebuttal: I thank the reviewers for clarifying my doubts and incorporating the suggestions into the paper. I've raised my rating to 8.

---

> ### Author Response · Authors · 2022-11-10
> **Response to Reviewer 9KJi**
>
> Thank you for your review! Thank you also for the suggestions to improve the exposition—we have implemented them in the updated draft (changes are in brown).
>
> Your central concerns seem to be the following:
>
> > *"“The poor algorithmic efficiency of optimization algorithms for NCE due to an exponentially flat Hessian of the loss was discussed in Liu et al. [1]. Therefore, the novelty of the paper appears to be limited.”"*
>
> Liu et al. uses flatness of the Hessian to prove lower bounds on first-order methods like gradient descent, while this paper addresses a more fundamental issue of the sample complexity. A sample complexity lower bound implies that without a large number of samples, we will not learn the true distribution, **no matter what optimization algorithm** is used--even one which overcomes the challenge of exponentially flat Hessian.
> Although the flatness of the Hessian is an ingredient in our proof, it does not directly control the sample complexity. Instead, we have to control the covariance matrix of the NCE loss, and a substantial amount of technical machinery needs to be developed to handle this. We’ve updated the draft to clarify this.
>
> ---
>
> > *"“The relevance of the novel results for the use of eNCE and normalized gradient method in Liu et al. [1] is not discussed.”"*
>
> While we build upon some of the observations in Liu et al. (the flatness of the NCE loss) for some of our proofs, the remaining contributions in Liu et al. are orthogonal to our work. Namely, the eNCE loss is a *different loss* from NCE—so its statistical complexity may be very different from NCE (neither our work, nor Liu et al address this). Our goal is to design a (relatively simple!) distribution that is difficult to learn using *NCE*, not a distribution that’s difficult to learn using *any algorithm*. Furthermore, the normalized gradient descent proposed in Liu et al only addressed the algorithmic difficulty stemming from the flat landscape—it’s entirely possible that there are statistical obstructions similar to the ones we point out in our paper.
>
> ---
>
> > *"“The effect of utilizing sample mean and covariance instead of the population mean, and covariance is not discussed.”"*
>
> Using population mean and variance only helps us, since the resulting noise distribution is closer to $p^*$ than the Gaussian with matching sample mean and covariance. In other words, a hardness result for NCE under known mean and covariance (which we prove) is stronger than a hardness result where the mean and covariance must be estimated from data.
>
> ---
>
> We hope that this clarifies things, and we would be happy to answer any further questions that you have.

---

### Official Review · Reviewer_mqhY · 2022-10-25

**Confidence:** 3
**Correctness:** 4
**Technical Novelty And Significance:** 3
**Empirical Novelty And Significance:** 2
**Recommendation:** 5

**Clarity, Quality, Novelty And Reproducibility:**

The main results in Theorems 3 and 4 are stated clearly. However, I don't find enough explanation on why such theorems show the main message of this paper, i.e., the pitfalls of Gaussians as a noise distribution in NCE. Although the results in Theorem 3 and 4 show the problems in NCE under this setup, there is no reason to believe those problems are all due to the Gaussian noise distribution. In fact, I suspect that if Gaussian noise distribution is replaced by some other simple distributions (e.g., uniform distribution), such problems may still appear. If the authors want to convey the message as the title of this paper, then the authors should at least provide evidence that other distributions do not have the problems stated in Theorems 3 and 4.

Some minor comments:
1. What is the precise definition of $E_{\theta}(x)$?
2. The proofs take a lot of space of the main body of this paper. I felt lost when reading the details in those proofs. I would suggest only highlighting the technique contribution (i.e., the special parts) of the proof and put the rest to the appendix to avoid distraction.

**Strength And Weaknesses:**

Strength: The results and the proof seem solid.
Weakness: The explanation of the results is not enough. The main message of this paper is questionable. See my comments below.

**Summary Of The Paper:**

This paper studies the noise contrastive estimation (NCE). NCE aims to learn probability density functions by first choosing a simple "noise" distribution and then training the parameters by minimizing the NCE loss, where the NCE loss measures the difference between the target distribution and the "noise" distribution. The authors show that when the target distribution is in a product form and the chosen "noise" distribution is Gaussian, the Hessian of the loss function decreases exponentially fast when the input dimension increases. In this situation, the authors also show that when the number of training samples goes to infinity, the mean-squared error is on the order of $\exp(d)/n$.

**Summary Of The Review:**

The setup and the results of this paper are stated clearly. However, I am skeptical about the message that this paper is trying to convey.

---

> ### Author Response · Authors · 2022-11-10
> **Response to Reviewer mqhY**
>
> Thank you for the review! We address your concerns in the following:
>
> ---
>
> > *"there is no reason to believe those problems are all due to the Gaussian noise distribution. In fact, I suspect that if Gaussian noise distribution is replaced by some other simple distributions (e.g., uniform distribution), such problems may still appear.”*
>
> You are correct, NCE might be statistically poorly behaved with other noise distributions as well!
> The reason we specifically focus on the Gaussian is that it is the canonical noise distribution chosen in practice, because it is: (1) easy to fit from the training data; (2) easy to sample from; (3) has an explicit expression for the density. Our proof only essentially uses the fact that $q$ is a product distribution, and that the coordinatewise marginals of $p^*$ and $q$ are at some constant total variational distance away from each other.
>
> The takeaway point of the paper isn’t that the Gaussian is a uniquely bad noise distribution: rather it is that distances between distributions tend to be amplified in high dimensions in a way that is detrimental to NCE, so that the noise distribution likely needs to be more carefully selected in high dimensions, in order for NCE to have good statistical efficiency.
>
> ---
>
> >*“What is the precise definition of $E_\theta(x)$?”*
>
> $E_\theta(x)$ is any function of $x$ parameterized by $\theta$ that characterizes the family of distributions of the form $p_{\theta}(x) \propto \exp(E_\theta(x))$ that we are trying to learn via NCE. For example, in our paper, we consider the setting where $E_{\theta}(x) = \theta^T T(x)$ where $T(x) = (x_1^4, \dots, x_d^4)$.
>
> ---
>
> We hope that this clarifies things, and we would be happy to answer any further questions that you have.

---

> ### Author Response · Authors · 2022-11-18
> **Response to Reviewer mqhY**
>
> We thank you again for your review !
> We hope that our response clarified that our results do indeed apply to much more general settings of the noise distribution—the reason for our focus on the Gaussian was solely because of its status as the "canonical" noise distribution in practice.
> Since the discussion period is coming to a close today, please let us know if there are any other questions we can answer.

---

### Official Review · Reviewer_ZzEv · 2022-10-25

**Confidence:** 3
**Correctness:** 3
**Technical Novelty And Significance:** 3
**Empirical Novelty And Significance:** 3
**Recommendation:** 8

**Clarity, Quality, Novelty And Reproducibility:**

This work is clear, seems novel and reproducible. I am not extremely familiar with the NCE literature, but if the Gaussian distribution is indeed the vastly most used noise distribution I think that the contribution made by this paper is definitely worth publishing.

Minor comments regarding the writing:
- Thm 3: $\Omega(d)$ is not defined
- Sec 3.1: the gradient and the Hessian **of** the NCE
- Eq 3-8: $L$ is applied indifferently to the distribution or the density
- Eq 8: I don't think that $TV$ has been introduced
- Lem 5: the existence of $\hat{P}$ is not proven yet. It is a bit weird to delay it to the bottom of page 4
- Lem 5: the Hellinger distance is not defined
- page 4: Since... I would use $\partial$ instead of $d$, that is already the dimension
- Sec 3.4: $\Omega(d)$ should be defined

**Strength And Weaknesses:**

**Strengths**
- The paper is globally clear and well written
- It tackles an interesting problem and gives an elegant solution

**Weaknesses**
- After proving the negative result about the Gaussian distribution, I would have liked the authors to elaborate a bit more on which distribution should be used instead. In particular, the authors mentioned two works in the conclusion, that learn $Q$ simultaneously as solving the NCE, but would any fixed distribution suffer from the same problem as the Gaussian?
- Some aspects of the writing might be improved, see below

**Summary Of The Paper:**

This paper shows that using the Gaussian distribution with mean and covariance of the data as the noise distribution in Noise Contrastive Estimation might be bad in terms of both statistical and computational efficiency.

**Summary Of The Review:**

Overall I like the paper and think it should be accepted

---

> ### Author Response · Authors · 2022-11-10
> **Response to Reviewer ZzEv**
>
> Thanks you for your review! We are glad you liked our work! We have revised our submission, corrected the typos and implemented your writing suggestions (the edits are in brown). Regarding your other concern:
>
> > *"but would any fixed distribution suffer from the same problem as the Gaussian?"*
>
> It may be true that any fixed noise distribution suffers from the same problem as the Gaussian (in the sense that there exists some family for which NCE performs poorly). We don’t know what is the most general result that we could state, but we believe we can generalize our result as follows: if the distributions $p*$ and $q$ are product distributions, and the coordinatewise marginals of $p^*$ and $q$ are at some constant total variation distance away from each other, the same problems persist.
>
> ---
>
> We hope that this clarifies things, and we would be happy to answer any further questions that you have.

---

> > ### Comment · Reviewer_ZzEv · 2022-11-21
> > **Post rebuttal**
> >
> > I thank the authors for their feedback. After having read other reviews and answers I keep my 8 score.

---

### Decision · Program_Chairs · 2023-01-20

**Decision:**

Accept: poster

**Justification For Why Not Higher Score:**

While this is a very nice paper I did not see any particular reason for a highlight. But would not be against either.



**Justification For Why Not Lower Score:**

No reason to reject.

**Metareview: Summary, Strengths And Weaknesses:**

Reviewers overall agree this paper should be accepted. I think the reviews summarize very well the strengths and weaknesses of the paper as well as points that the authors should include in the revised version. It is important to publish also negative results on broadly used methods and this paper is a good addition to that literature.

**Note From Pc:**

if the above contains the word "oral" or "spotlight" please see: "oral" presentation means -> notable-top-5% and "spotlight" means -> notable-top-25%. As stated in our emails, we are disassociating presentation type from AC recommendations